# CaSR Gene Polymorphisms and PHPT Phenotypes: What Else Can We Learn? A Single-Center Experience on a Cohort of Italian Patients

**DOI:** 10.3390/genes16080974

**Published:** 2025-08-19

**Authors:** Michele Cannito, Giacomo Voltan, Giulia Carraro, Michela Ferrarese, Giacomo Contini, Carlo Mogno, Loris Bertazza, Susi Barollo, Francesca Torresan, Maurizio Iacobone, Caterina Mian, Valentina Camozzi

**Affiliations:** 1Endocrinology Unit, Department of Medicine (DIMED), Via Ospedale Civile 105, 35128 Padua, Italy; michele.cannito@studenti.unipd.it (M.C.); michela.ferrarese.1@studenti.unipd.it (M.F.); giacomo.contini@aopd.veneto.it (G.C.); carlo.mogno@aopd.veneto.it (C.M.); loris.bertazza@unipd.it (L.B.); susi.barollo@unipd.it (S.B.); catrina.mian@unipd.it (C.M.); valentina.camozzi@aopd.veneto.it (V.C.); 2Endocrinology Unit, University-Hospital of Padova, Via Giustiniani 2, 35128 Padua, Italy; giulia.carraro@aulss6.veneto.it; 3Endocrine Surgery Unit, Department of Medicine (DIMED), 35128 Padua, Italy; francesca.torresan_01@aopd.veneto.it (F.T.);; 4Endocrine Surgery Unit, Department of Surgical, Oncological and Gastroenterological Sciences DISCOG, 35128 Padua, Italy

**Keywords:** Primary Hyperparathyroidism (PHPT), Calcium-Sensing Receptor (CASR), Single Nucleotide Polymorphism (SNPs), hypercalciuria

## Abstract

**Purpose:** This study investigates the role of CASR gene polymorphisms (A986S, R990G, Q1011E) in PHPT genetic susceptibility and its clinical variability. The aim is to evaluate the prevalence of these polymorphisms in patients with sporadic PHPT and their impact on clinical course, biochemistry, and histological features. **Methods:** 106 patients underwent clinical and anamnestic evaluations, focusing on major PHPT complications, as well as biochemical analyses of blood and urine. Genetic testing was conducted for CASR gene polymorphisms. Histological data were available for 68 patients who underwent parathyroidectomy. **Results:** The sample included 83 women and 23 men; mean age at diagnosis was 54.5 years. 55 patients carried CASR gene polymorphisms, while 51 were wild-type. Prevalence rates of polymorphisms were consistent with data for the Caucasian population, with A986S being the most common (31%). No significant associations were found between polymorphisms and increased levels of ionized calcium or other blood phospho-calcium metabolism parameters. However, 24-h urinary calcium levels were higher in patients with polymorphisms (*p* = 0.0185), particularly in those older than 50 years (*p* = 0.030) and with the A986S variant. Hypercalciuria was predictive of CASR polymorphism presence (OR = 2.76, *p* = 0.003). No significant association with PHPT complications, such as renal calculi or bone involvement, was confirmed. Histological data revealed no clear links between polymorphisms and more aggressive variants. **Conclusions:** CASR gene polymorphisms are associated with hypercalciuria but do not significantly influence age of onset or clinical phenotype in PHPT. Genetic analysis may be useful in selected cases to better understand individual clinical profiles.

## 1. Introduction

Primary hyperparathyroidism (PHPT) is the third most common endocrine disease, after diabetes and thyroid diseases [1], resulting from the excessive and autonomous production of parathyroid hormone by a single parathyroid adenoma in 80–85% of cases, in 15–20% by diffuse hyperplasia (sometimes asymmetrical), and less than 1% by a parathyroid carcinoma [2]. PHTP is typically characterized by the combination of hypercalcemia and elevated, or inappropriately high, PTH levels. In addition, subjects with PHPT usually present low serum phosphate and increased urinary calcium excretion [3]. This might lead to many comorbidities due to the increase in blood calcium level, such as muscular, gastrointestinal, neurological and cardiovascular disorders [4], reduced bone density, fragility fractures [5,6], and nephrolithiasis which can progress to renal failure [7]. PHPT encompasses a wide spectrum of clinical phenotypes, with the Calcium-Sensing Receptor (CASR) being one of the possible candidates to explain genetic susceptibility to PHPT and its clinical phenotypes [8]. CASR is a ubiquitous membrane receptor that detects changes in extracellular calcium concentration, controlling PTH secretion in the parathyroid glands and calcium reabsorption in the renal tubule, to achieve homeostasis [9,10].

CASR is highly expressed in the parathyroid glands: elevated calcium levels activate the receptor, inhibiting PTH secretion; on the contrary, decreasing blood calcium level inactivates the CASR receptor, stimulating PTH secretion within a few seconds, an increase in PTH synthesis within minutes and, in the case of chronic hypocalcaemia, parathyroid hyperplasia [11,12].

In PHPT, higher calcium concentrations are found, with accelerated calcium reabsorption in bones, kidney and intestine as well [13,14].

At the renal level, CASR is thought to play a crucial physiological role in the defend against hypercalcaemia, promoting urinary calcium excretion independently of PTH action [15]. In addition, it may also exert an anti-phosphaturic effect by reducing circulating PTH [16].

Despite several mutations have been described in CASR gene [17], ranging from a loss of function of the receptor, like those that affect subjects with familial hypocalciuric hypercalcemia and severe neonatal hyperparathyroidism, to a gain of function in the case of autosomal dominant hypocalcaemia [18].

The most common genetic variants are due to the substitution of a single nucleotide, known as “single nucleotide polymorphisms” (SNPs). One SNP in coding and regulatory regions of genes, might influence gene functions, biological processes and cellular activities. These are called “susceptibility polymorphisms”, since they increase the risk of diseases without being a direct cause [19]. Within the human CASR gene, several polymorphisms have been described [18] that could alter the function of this receptor.

The relationship of these polymorphisms with calcium and PTH levels has been extensively studied in healthy populations and in patients with calcium metabolism disorders. The results of the studies conducted so far are not conclusive and often have led to discrepant results, which could depend on the differences in the genetic background of the groups studied [20,21,22,23,24,25,26,27,28,29,30,31,32,33,34,35].

Hence, the aims of the present study are to evaluate the prevalence of polymorphisms A986S, R990G and Q1011E of the CASR gene in a cohort of Italian patients with sporadic PHPT, to determine the impact of CASR gene polymorphisms on metabolism phosphor—calcium and the clinical course of the disease, to describe the association between CASR gene polymorphisms, and the histological features in our cohort.

## 2. Materials and Methods

### 2.1. Patients’ Selection

In this retrospective, monocentric study, we considered 168 Italian patients, admitted to our Endocrinology Unit of the Padua Hospital between 2020 and 2024, on whom genetic analysis for CASR, MEN1 e CDC73 genes was performed during the process of diagnosis and therapeutical management of bone-related conditions. The following inclusion criteria were applied: (a) patients affected by PHPT; (b) patients who provided written informed consent to genetic testing; (c) patients presenting CASR gene polymorphisms (A986S, R990G, Q1011E). PHPT diagnosis was based on high levels of ionized calcium in the presence of high serum PTH levels, with adequate levels of 25-OH vitamin D and normal renal function. Pre-surgical identification of parathyroid lesions was performed through ultrasound examination of the neck and/or scintigraphy with technetium-sestamibi, and in some cases with 18Fluorocholine-PET/RM. In patients undergoing parathyroidectomy surgery, anatomopathological examination was also performed on the surgical sample. The following exclusion criteria were applied: (a) patients affected by other diseases such as FHH, hypoparathyroidism, MEN syndromes, CDC73-related disorders and their first-degree relatives; (b) patients with SNPs of MEN1 and CDC73 genes; (c) patients who presented polymorphisms of the CASR gene different from those under study; (d) patients with other comorbidities interfering with phospho-calcium metabolism or that were already taking interfering drugs (thiazide diuretics, bisphosphonate, cinacalcet); (e) patients with incomplete medical records. According to our inclusion and exclusion criteria, a total of 106 subjects were finally selected. They were subsequently split into two groups based on the presence/absence of A986S, R990G, Q1011E polymorphisms of the CASR gene in their genetic analysis, as resumed in Figure 1.

### 2.2. Anamnesis

Demographic data were collected, including patient’s age at diagnosis, sex and geographical origin. A detailed anamnestic history was collected for all patients, with particular attention to the search for nephrolithiasis (defined as a history of renal colic with expulsion or identification by ultrasound imaging) and bone complications (osteopenia/osteoporosis via DEXA and/or clinical or radiologically documented fragility fractures). It was administered a questionnaire to evaluate patients’ dietary calcium intake [36], aimed to determine whether an excessive amount of calcium excreted in the urine could potentially be attributed to a high dietary intake.

### 2.3. Biochemistry and Genetic Analysis

Biochemical parameters related to phospho—calcium metabolism were reported, such as blood calcium, total albumin, ionized calcium, blood phosphate, PTH, 24-h urinary calcium and 25 (OH)-D. The measurement of these analytes was performed using standard commercially available laboratory kits. The PTH tests were conducted across various laboratories with differing generations of tests, having varying upper limits (37 pg/mL for third generation tests and 65–80 pg/mL for older ones); taking into account these differences, the upper limit of each test was used as a reference, and comparisons between patients were made using the ratio of the measured PTH concentration to the respective upper limit norm.

The three most frequent single nucleotide polymorphisms (SNPs) map on the portion of exon 7 that encodes the intracellular tail of CASR, and are: A986S, characterized by the change of codon 986 from GCC to TCC causes an alanine/serine amino acid substitution; R990G, with the change of codon 990 from AGG to GGG causing an arginine/glycine amino acid substitution; Q1011E, with the change of codon 1011 from CAG to GAG causing a glutamine/glutamate amino acid substitution. DNA extraction from blood was performed using the protocol of the QIAmp DNA tissue mini-Kit (Qiagen, Hilden, Germany). The purification procedure includes 4 steps: lysis of the biological sample, absorption/binding of DNA to the silicon membrane contained in the QIAamp column, washing and removal of contaminating residues, DNA elution. After purification, the sample is subjected to polymerase chain reaction (PCR). This technique allows to selectively amplify specific DNA sequences or in a complex population of cDNA. Exons 1, 2, 3, 4, 5 and 6 of the CASR gene were amplified by PCR. Sequencing PCR of the purified product is performed following the Big-dye Terminator protocol. The sample obtained from the PCR is subsequently verified by means of a horizontal electrophoresis on a 2% (W/V) agarose gel in TAE (50X). At the end of this procedure, the product is purified from salts, dNTPs and low molecular weight molecules using AutoSeq G-50 Dye columns Terminator Removal Kit (GE Healthcare, Chicago, IL, USA,). The product of this purification is then processed on the automatic sequencer ABI PRISM (Applied Biosystems, Waltham, MA, USA) to determine the sequences of the exons of the CASR gene.

### 2.4. Statistical Analysis

The data of the study subjects were collected and entered in a database created with Microsoft Excel^®^ version 365 and subsequently processed using the statistical software MedCalc^®^ version 20.2 (MedCalc Software Ltd., Ostend, Belgium; https://www.medcalc.org; 2022, accessed on 1 December 2024). Continuous variables were expressed as mean ± standard deviation, if parametric or as interquartile range if non-parametric. Categorical variables were presented as number and frequency in percentages. The comparison between groups of continuous variables was performed with the Mann Whitney test for non-parametric variables and with the student T test for parametric ones. ANOVA test was utilized to compare three or more groups. The comparison between categorical variables was performed using the Fisher exact test or the chi-square test. A value of *p* < 0.05 was considered statistically significant.

All data analyzed during this study are securely stored in the data repositories of the University of Padova—Reasearch Data UniPD [37].

## 3. Results

### 3.1. Study Population

Our sample was composed by 106 Italian subjects affected by PHPT. The demographic and clinical data of the patients are reported in Table 1. Genetic analysis of the polymorphisms A986S, R990G and Q1011E was performed: in 55 patients the presence of at least one of these variants was found, while 51 were negative. Among the 52% of patients with CASR gene SNPs, the majority (31%) carried the A986S variant. A small percentage (8%) carried the Q1011E and R990G variants; 5 patients in total carried two variants (2 R+Q and 3 A+Q). Out of the whole population, 68 patients underwent surgical parathyroidectomy, 22 were waiting to receive the surgery, for 16 surgery was not recommended or patient refused it. Indication for surgery were upheld according to current guidelines [38]. Data relating to the histological diagnosis of the operated patients, available for 68 out of 68 patients are reported in Table 2.

### 3.2. Correlation Between CASR Gene Polymorphisms and Clinical Characteristics of Subjects with PHPT

The sample was subsequently divided into two subgroups based on the presence/absence of CASR gene polymorphisms, to compare the demographic, biochemical and clinical characteristics of the two groups (see Table 1).

No statistically significant differences were found between the two groups in terms of sex and mean age at the time of PHPT diagnosis. The analysis of the parameters of phospho-calcium metabolism was also negative for any statistically significant difference among the two groups, except for 24 h urinary calcium levels, that resulted higher in subject tested positive for CASR polymorphism (9.9 ± 3.6 mmol/24 h vs. 8.2 ± 4.5 mmol/24 h, *p* = 0.0185).

The major complications of primary hyperparathyroidism, namely skeletal manifestations (presence of osteopenia or osteoporosis) and nephrolithiasis, were also evaluated: patients with and without CASR gene polymorphisms didn’t show a significant difference in their prevalence.

### 3.3. Further Sub-Analyses Conducted on 24-h Urinary Calcium Values

As already described and reported in Table 1, median values of calcium excreted in the urine over 24 h were higher in subjects carrying CASR polymorphism than in non-carriers, with statistically significant differences (see also Figure 2).

We then proceeded to stratify the patients based on the age at diagnosis, identifying three groups: (a) patients with an age at diagnosis of less than 35 years (8 patients); (b) patients with an age at diagnosis between 35 and 50 years (39 patients); (c) patients with age at diagnosis greater than 50 years (59 patients). In group (a) no statistically significant difference was found (*p* = 0.527); similar result for group (b) (*p* = 0.555). In group (c) the data of the entire population is confirmed, where therefore the urinary calcium value was significantly higher in subjects carrying the polymorphism (*p* = 0.030) (Figure 3).

Patients were stratified into three subgroups according to the type of polymorphism, and separate analyses were performed for each. A weak positive correlation between blood and urinary calcium was observed only in A986S carriers (*p* = 0.047), but not in Q1011E or R990G carriers. In the entire cohort (*n* = 106), univariate logistic regression showed that each 0.25 mmol/24 h increase in urinary calcium was associated with a higher likelihood of carrying a polymorphism (OR = 1.29; *p* = 0.016). The model remained significant when using the categorical variable “hypercalciuria” (>7.5 mmol/24 h) (OR = 2.76; *p* = 0.003; Figure 4).

As a final analysis, a correlation was sought between the presence of hypercalciuria and the clinical and/or radiological evidence of kidney stones: no statistical association was found. Indeed, in terms of sample size, there were more patients with kidney stones in subjects with high levels of renal calcium excretion.

### 3.4. Correlation Between CASR Gene SNPs and Age at Diagnosis

Our study aimed to investigate whether a SNP increased susceptibility to developing PHPT (thus determining its onset at earlier age) Initially, patients were divided into two groups—under 50 and over 50 years—based on age threshold for surgical management of PHPT, but no significant difference in polymorphism prevalence was found (*p* = 1). Further analysis were performed dividing the sample into four age subgroups (<35, 35–50, 50–65, >65 years), but no higher prevalence of positive genetic tests for polymorphism was identified in younger age groups using the Chi-squared test.

### 3.5. Correlation Between CASR Gene SNPs and Complications in Target Organs

For skeletal damage, 30 patients without SNP and 36 carrying one SNP showed signs of involvement, while no bone issues were found in 8 patients without SNP and 17 with the polymorphism: statistical analysis showed no significant difference (*p* = 0.248). Regarding renal calculosis, it was present in 15 patients without SNP and 18 with the polymorphism, while absent in 34 patients without SNP and 35 with the polymorphism: again, no significant difference was observed (*p* = 0.719).

### 3.6. Correlation Between CASR Gene SNPs and Histology

The results of the anatomopathological examination are reported in Table 2. Dividing patients into two groups based on the presence or absence of CASR polymorphism (38 with SNPs and 30 wild-type), the SNP group showed higher frequencies of atypical adenomas and parathyroid carcinomas, whereas typical adenomas were more prevalent in the wild-type group: however, chi-square analysis did not reveal statistically significant differences (*p* = 0.1796). Further grouping into two categories—simple adenomas versus atypical adenomas and carcinomas—yielded borderline statistical significance using the Fisher test (*p* = 0.0421) (see Figure 5). Considering only the 38 patients with CASR gene polymorphism, the type of polymorphism found according to the various types of histological diagnosis is also reported in Table 2. A prevalence analysis was conducted, this time according to the type of polymorphism found with the genetic analysis, which did not give significant results.

## 4. Discussion

The calcium-sensing receptor plays a pivotal role in maintaining calcium homeostasis in the body, acting as a sensor for plasma calcium levels and thus regulating PTH secretion by parathyroid glands; it also controls calcium reabsorption in the kidneys, and is expressed in many other tissues, such as bones and intestines, highlighting its extensive influence on calcium regulation [10]. Mutations in the CASR gene can lead to significant alterations in calcium levels. Consequently, the CASR gene may be considered a key factor in explaining individual variations in serum calcium levels, both in healthy individuals and in hypercalcaemic patients.

Starting from these considerations, over the past two decades, research has focused on three single nucleotide polymorphisms (SNPs)—A986S, R990G, and Q1011E—affecting exon 7 within the CASR gene, which encodes the intracellular portion of the receptor: it has been hypothesized that those could influence calcium concentrations in healthy populations. The first work supporting this hypothesis was published in 2001 by Cole et al. who linked the A986S polymorphism to higher calcium levels in a population of healthy Canadian women [20]. However, subsequent studies have yielded inconsistent and conflicting results. Furthermore, a still debated issue is whether the effect of these associations is significant enough to impact on clinical conditions related to phospho—calcium metabolism imbalances, such as sporadic primary hyperparathyroidism. Research in this area is still relatively sparse and inconclusive. It should be emphasized that in multifactorial diseases, it’s challenging to quantify the role of these polymorphisms to disease susceptibility, due to the complex interplay of genetic and environmental factors influencing the final resulting phenotype.

In Caucasians, the A986S variant is the most common, occurring in about 30% of healthy subjects and approximately 40% in subjects with PHPT [29]. On the other hand, the R990G and Q1011E variants are less common in both healthy individuals and subjects with PHPT but are more prevalent respectively in the Asian (R990G) [23] and African (Q1011E) population. Our findings align with these estimates, since among the 55 patients with one CASR gene polymorphism, the majority (31%) carried the A986S variant, whereas R990G and Q1011E presence, presence was found, as expected, in a minor—albeit significant—percentage of patients (8% of total). Finally, 5 patients in total carried two variants.

According to the presence or absence of SNP, no statistically significant differences were observed in terms of sex and the average age at the time of PHPT diagnosis.

Several studies [20,22,25,29] reported that subjects carrying polymorphisms showed higher values of albumin—corrected calcium, PTH, and ionized calcium. Our study diverges with these findings as these associations were not found to be statistically significant. Similarly, this lack of difference was also confirmed by separately evaluating the impact of the three individual types of polymorphism: these were not associated with higher concentrations of ionized and total corrected calcium. Miedlich et al. [22], in a cohort of German patients, and by Scillitani et al. [29] in a cohort of Italian patients, observed that the Q1011E variant was associated with higher levels of PTH and lower levels of serum phosphate, differently from our findings, which did not highlight this evidence.

The only significant data concerned the 24-h urinary calcium, which resulted higher in subject carrying CASR polymorphism (*p* = 0.0185). This is a finding already reported in the literature by Corbetta et al. [30] and Vezzoli et al. [25] respectively on women with PHPT and on healthy women with kidney stones, respectively. Interestigly, this observation was confirmed only in patients over 50 years (*p* = 0.030), when stratifying according to the age at diagnosis, probably due to the higher sample size compared to the younger group of patients.

Hence, the separate analysis of each type of polymorphism showed as the only statistically significant finding a weak positive correlation between blood calcium levels and urinary calcium levels in patients with the A986S polymorphism (*p* = 0.047): this indicates that as blood calcium levels rise, there is a corresponding increase in urinary calcium excretion. This correlation was not observed in patients with the Q1011E and R990G polymorphisms, likely due to the small sample size of eight subjects. Moreover, this confirms that the overall results are strongly sustained by the alterations found in patients carrying A986 SNP.

The 24-h urinary calcium data caught our attention because it is not unusual in clinical practice to encounter two patient phenotypes, one with PHPT who exhibit normocalciuria and others who exhibit hypercalciuria, irrespective of their blood calcium levels. Moreover, some patients respond favourably to thiazide treatment, while others do not. Understanding the underlying causes of these variations, including potential genetic factors, is of significant interest. Indeed, we might hypothesize that the presence of a CASR polymorphism in renal tubular cells could lead to increased calcium excretion; consequently, this may trigger hyperfunction of the parathyroid cells by feedback mechanism, resulting in hypersecretion of PTH aimed at calcium reabsorption to maintain calcemia within the normal range. This physiological response could potentially develop over time into chronic PHPT; however, our results do not show significantly higher PTH levels in patients with polymorphism, not fully confirming this hypothesis.

One of the most promising results comes from a univariate logistic regression, as for every 0.25 mmol/24 h increase in 24-h urinary calcium, the likelihood of having the polymorphism detected in genetic analysis increased byOdds ratio = 1.29, *p* = 0.016. Indeed, considering the categorical variable “hypercalciuria”, defined as a lab value above the cut-off of 7.5 mmol/24 h, the model’s significance was further confirmed (Odds ratio = 2.76, *p* = 0.003).

Considering the described pathophysiological mechanisms, our findings may carry relevant clinical implications: if confirmed in larger cohorts, the identification of specific CASR polymorphisms associated with distinct profiles could support a more personalized approach to the management of primary hyperparathyroidism, thus aiding clinicians in selecting patients who might benefit from targeted genetic testing. Notably, current international guidelines acknowledge the potential utility of genetic testing primarily for the differential diagnosis of familial or syndromic forms of PHPT; however, they do not offer specific recommendations regarding routine CASR genotyping in patients with apparently sporadic PHPT [38]. In contrast, experienced centers recommend NGS panel testing, including CASR, in patients who present certain features, aiming to identify cases with an elevated likelihood of underlying germline variants even in apparently non syndromic PHPT [39].

In this context, while we acknowledge the exploratory nature of our findings and the limited sample size, the observed associations may suggest that a subset of patients with apparently sporadic PHPT, who do not meet traditional criteria for genetic testing, could still benefit from CASR genotyping. Nevertheless, the routine adoption of such an approach must also contend with its currently unfavourable cost—effectiveness profile: in the absence of robust evidence linking CASR genotyping to improved patient—relevant outcomes or to a meaningful reduction in healthcare costs, indiscriminate testing risks imposing a significant economic burden without a proportionate clinical benefit.

As final analysis, we explored the correlation between hypercalciuria and the clinical and/or radiological evidence of renal calculosis: despite no statistical association was identified, we observed that a higher rate of patients with calculosis showed hypercalciuria. This could be secondary to the fact that renal calcium excretion is not the sole determinant for the formation of kidney stones: rather, the patient’s clinical phenotype is influenced by complex interactions within the complete urinary lithogenic profile, which includes, for example, the excretion of phosphorus, oxalates, citrate, cystine, and urinary pH.

In accordance with the existing literature, which does not report an earlier onset of PHPT in individuals carrying the polymorphism, our results showed no difference in terms of SNP presence and the age at diagnosis. Furthermore, Cetani et al. [27] even assert that the presence of the polymorphism does not increase the overall risk of developing PHPT.

We also investigated any association between the SNP presence and complications in target organs, specifically renal calculosis and skeletal involvement.

For skeletal involvement, in agreement with Cetani et al. [26], the Chi-squared test revealed no statistically significant differences between the two groups (*p* = 0.248).

Similarly, for renal involvement, the Chi-squared test did not detect statistically significant differences between the two groups (*p* = 0.719). Although the statistical analysis did not find significant associations, it is important to consider the clinical relevance of these findings. The lack of statistical significance could be attributed to a variety of factors, such as sample size or the heterogeneity of the study population: in other studies [32,33] that found a prevalence of nephrolithiasis in subjects carrying polymorphisms (particularly R990G), the comparison groups were not PHPT patients, but rather individuals with kidney stones and healthy subjects; therefore, the populations studied were different from our sample.

In 68 patients who underwent parathyroidectomy a deeper analysis considering the histological features was performed. Among these patients, the majority (32 patients) had typical adenomas. 38 patients showed polymorphisms while 30 were wild—type patients. A higher frequency of atypical adenomas and parathyroid carcinomas was observed in patients with polymorphisms, while typical adenomas were more common in those without polymorphisms, though, performing a Chi-square test for prevalence revealed no significant differences between the groups (*p* = 0.1796). However, when patients were grouped into those with simple adenomas and those with atypical adenomas or carcinomas, the Fisher test indicated a statistically significant (*p* = 0.0421).

According to several studies in the literature, CASR appears to influence the proliferation of parathyroid cells, playing a significant role in the development of both benign and malignant tumors [33]: its activation seems to exert an inhibitory effect on proliferation, while inactivating mutations stimulate parathyroid hyperplasia. Moreover, reduced expression of CASR has been demonstrated in parathyroid adenomas and, even more so, in parathyroid carcinomas with a high proliferative index.

The previous discussion applies to complete genetic mutations, which cause significant DNA changes leading to major alterations in protein function and can represent direct causes of certain pathologies. In contrast, our study focuses on susceptibility polymorphisms, subtle DNA variations that do not directly cause diseases but may increase the likelihood when combined with other risk factors: thus, the predictive significance of these variations is different. Further studies are therefore needed to obtain a decisive result.

A prevalence analysis was conducted only on the 38 patients with CASR gene polymorphism, categorizing them according to the specific polymorphism identified through genetic analysis: this prevalence analysis did not yield significant results concerning the type of lesion. Additionally, the small sample size prevented us from determining if any specific polymorphism was more strongly associated with parathyroid carcinoma.

Overall, in our cohort CASR polymorphisms were not associated with the presence of renal calculi, skeletal involvement, or aggressive histological features. This lack of statistically significant associations may suggest that these variants play a minimal role, if any, in determining PHPT severity. Nonetheless, this interpretation should be made with caution, as our study was limited by sample size. Albeit not consistently significant, we observed a trend toward a higher frequency of atypical adenomas and carcinomas in patients carrying polymorphisms, indicating that a possible clinically meaningful effect cannot be definitively ruled out. Further studies in larger and more homogeneous cohorts will be required to clarify whether CASR polymorphisms contribute to a more severe clinical phenotype or whether their influence is negligible due to the interplay of numerous other biological and environmental factors.

Due to its retrospective nature, this study has some limitations. A larger sample size would have enabled more significant conclusions regarding the association between polymorphisms and histology. Additionally, the lack of a long-term follow-up period prevented the evaluation of disease recurrence or persistence after parathyroidectomy in patients with and without polymorphisms. A medium-long term follow-up period would be useful to see who presents relapsing PHPT due to genetic predisposition, becoming hypercalciuric again.

Among the study’s strengths there are: the inclusion of younger age groups, the histologically confirmed diagnosis of PHPT in over half of the patients, and the method of evaluating the effect of a single polymorphism on the phenotype. Patients were divided based on the type of polymorphism present and then compared with wild-type subjects without any CASR polymorphism; this approach minimized interference from other polymorphisms. This decision was deemed appropriate since literature suggests that all three polymorphisms, despite different impacts and implications, are associated with alterations in the biochemical values of phospho—calcium metabolism and clinical parameters.

Prospects include the possibility of collecting a larger sample, to homogenize the results present in the literature, and to consider the confounding factors on the phenotype (age, sex, duration of disease, menopause, comorbidity, lithogenic profile) to be included in a multivariate analysis.

## 5. Conclusions

In the sample of patients in our study, positive findings include:
-The prevalence data of polymorphisms are consistent with what is known in the literature for the Caucasian population;-24-h urinary calcium values are higher in subjects carrying the polymorphism compared to wild type subjects (especially in those over 50 and in patients with the A986S polymorphism);-The presence of hypercalciuria can be predictive of CASR gene polymorphism; thus in selected cases, genetics may clarify the clinical picture.


Among other results:
-No association was found between the A986S, R990G and Q1011E polymorphisms of the CASR gene and increased levels of ionized calcium, nor other blood parameters of phospho-calcium metabolism;-A statistically significant prevalence of SNPs in younger age groups was not documented;-With reference to the complications of PHPT, the association between the R990G variant and renal calculosis was not confirmed; no association was found with bone involvement;-In patients with histological diagnosis, no statistically significant association was found between polymorphisms and more aggressive histological variants.


## Figures and Tables

**Figure 1 genes-16-00974-f001:**
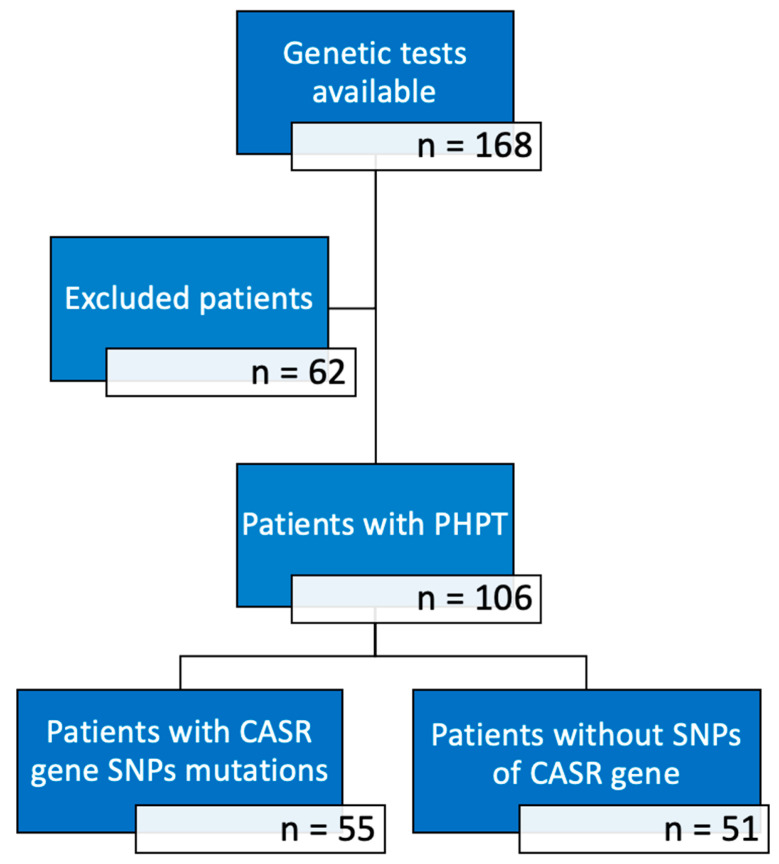
Number of subjects in the various phases of the study.

**Figure 2 genes-16-00974-f002:**
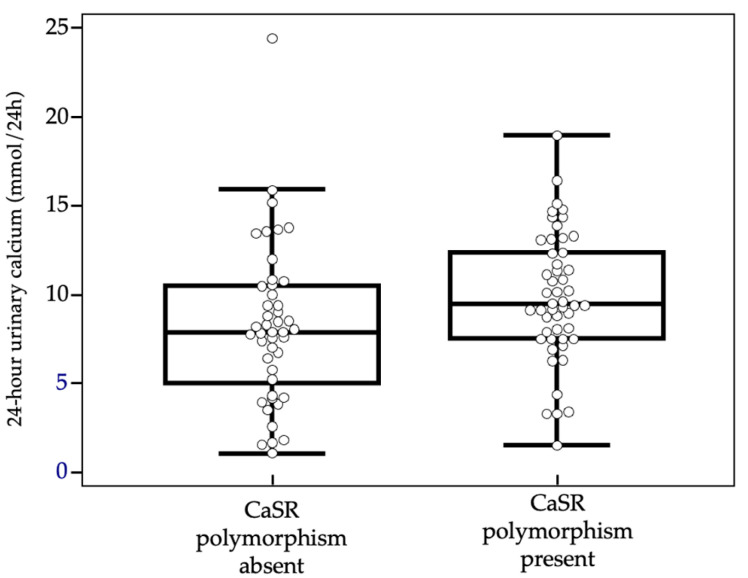
Urinary calcium values comparison with box-and-whiskers plot.

**Figure 3 genes-16-00974-f003:**
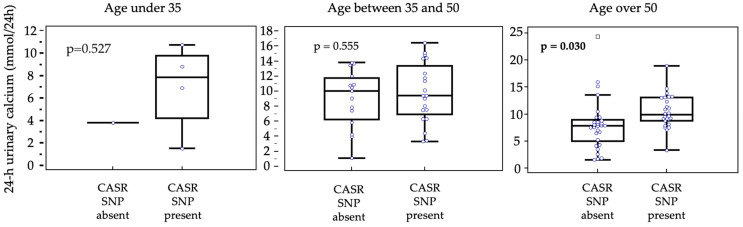
Urinary calcium values comparison with box-and-whiskers plot, according to patients’ age.

**Figure 4 genes-16-00974-f004:**
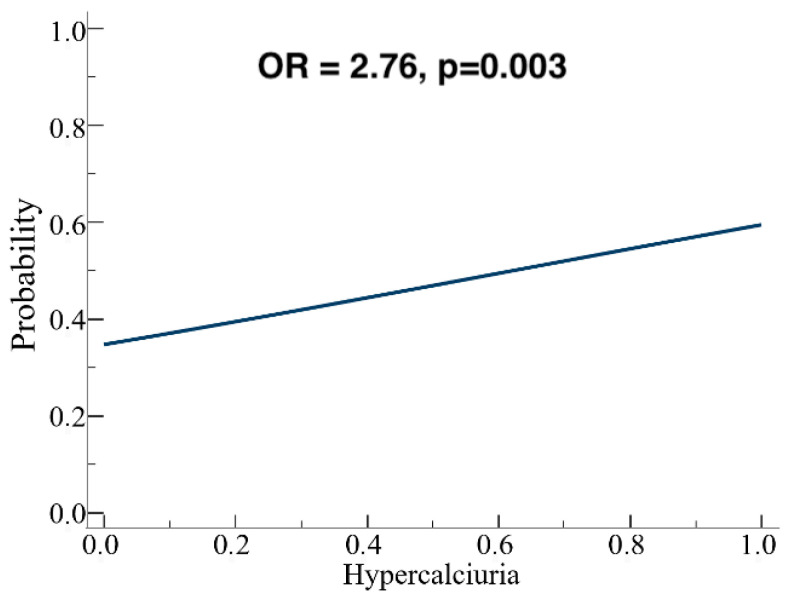
Logistic regression analysis: the presence of hypercalciuria is predictive for CASR gene SNPs presence.

**Figure 5 genes-16-00974-f005:**
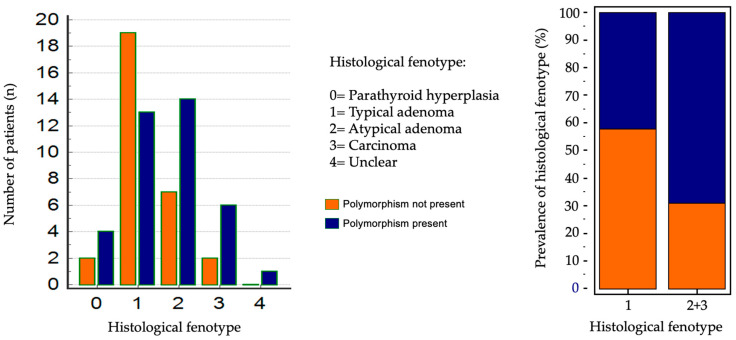
Statistical analysis of prevalence according to histological diagnosis.

**Table 1 genes-16-00974-t001:** General characteristics of the population examined, and comparison of subgroups of patients with and without CASR gene polymorphisms.

Parameters	Values	CASR Polymorphism		*p*-Value
		Present	Absent	
N° subjects	106	55	51	
Sex - Females - Males	84 (79%) 22 (21%)	46 (84%) 9 (16%)	38 (75%) 13 (25%)	
Age at diagnosis (years)	54.59 ± 13.58	53 ± 14	56 ± 14	ns
Blood calcium (corrected for albumin) (mmol/L)	2.88 ± 0.47	2.9 ± 0.37	2.9 ± 0.56	ns
Ionized calcium (mmol/L)	1.40 ± 0.13	1.4 ± 0.08	1.4 ± 0.18	ns
Phosphorus (mmol/L)	0.81 ± 0.21	0.81 ± 0.18	0.81 ± 0.23	ns
PTH/ULN-PTH ratio	2.68 ± 2.92	2.17± 0.99	3.09± 3.96	ns
24 h urinary calcium (mmol/24 h)	9.14 ± 4.08	9.9 ± 3.6	8.2 ± 4.5	0.0185
PHPT complications - Kidney stones - Skeletal involvement	33 (31%) 66 (62%)	18 (33%) 36 (65%)	15 (29%) 30 (59%)	ns
N° patients who underwent surgery	68 (65%)	38 (67%)	30 (59%)	

PTH: Parathyroid hormone; ULN: Upper Limit of norm; PHTPT: Primary Hyperparathyroidism, ns: not significant, N°: number

**Table 2 genes-16-00974-t002:** Histological diagnosis in patients who underwent parathyroidectomy surgery, and type of polymorphism found in each group.

Histology	Numerosity	CASR Polymorphism	
		Present	Absent
Hyperplasia	6	4 (2 A986S; 2 R990G)	2
Typical Adenoma	32	13 (7 A986S; 2 Q1011E; 1 R990G; 2 A+Q; 1 R+Q)	19
Atypical Adenoma	21	14 (9 A986S; 3 Q1011E; 2 R990G)	7
Carcinoma	8	6 (3 A986S; 3 Q1011E)	2
Not clear	1	1 (A986S)	-

## Data Availability

All data generated or analyzed during this study are included in this published article.

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
