# Peer review of "CaSR Gene Polymorphisms and PHPT Phenotypes: What Else Can We Learn? A Single-Center Experience on a Cohort of Italian Patients"

_genes, 2025, doi:10.3390/genes16080974_

Round 1

Reviewer 1 Report

Comments and Suggestions for Authors

The authors reported a single-center experience of CaSR gene polymorphisms and PHPT phenotypes.

This study provides a valuable investigation into the potential role of CASR gene polymorphisms in primary hyperparathyroidism, assessing their impact on clinical, biochemical, and histological features. The findings contribute to the ongoing discussion about genetic susceptibility in PHPT and whether certain CASR variants influence disease severity or complications.

Here are my comments :

  • Adjust references in intro ; put in same brackets multiple refs
  • No need in intro to explain the polymorphisms : let them in Methods
  • In methods explain the diagnosis of PHPT : levels of calcemia, PTH and phosph
  • Was MEN 2 excluded ?
  • In tables : put points not commas
  • Results are well presented , somehow long. some sentences are summarized in tables and figures and could be deleted.
  • Put limitations / strengths at the end of discussion.
  • clinical implications must be added , correlated with current guidelines.
  • Add a cost limitation of the procedure.
  • Hypercalciuria was high in patients over 50 yo : this suggest an age-related modulation of CASR function, or  due to longer disease exposure?
  • The study did not find a link between CASR polymorphisms and renal calculi, bone disease, or aggressive histology. Please better explain in discussion if this imply that these variants have a minimal role in PHPT severity, or was the sample size too small ?
  • Are there differences in response to medical therapy : cinacalcet; based on CASR genotype?
  • references are not numbered in the end.

Author Response

The authors reported a single-center experience of CaSR gene polymorphisms and PHPT phenotypes.

This study provides a valuable investigation into the potential role of CASR gene polymorphisms in primary hyperparathyroidism, assessing their impact on clinical, biochemical, and histological features. The findings contribute to the ongoing discussion about genetic susceptibility in PHPT and whether certain CASR variants influence disease severity or complications.

Adjust references in intro ; put in same brackets multiple refs

  • Thank you for the suggestion. We have revised the introduction accordingly, grouping multiple references in the same brackets where appropriate

No need in intro to explain the polymorphisms: let them in Methods

  • Thank you for your suggestion. We have moved the detailed explanation of the polymorphisms from the Introduction to the Methods section as requested, leaving a brief sentence in the Introduction to facilitate reader understanding.

In methods explain the diagnosis of PHPT: levels of calcemia, PTH and phosph

  • Thank you for your comment. The diagnostic criteria for PHPT, including serum calcium, PTH, and phosphate levels, were already detailed in the Methods section. For your convenience, we highlight the relevant passage below:

lines 93-94: “PHPT diagnosis was based on high levels of ionized calcium in the presence of high serum PTH levels, with adequate levels of 25-OH vitamin D and normal renal function.”

Was MEN 2 excluded?

  • We sincerely thank you for this pertinent and thoughtful comment. Genetic testing for MEN2 was not systematically performed in our cohort. This type of analysis is reserved to patients carrying a personal or familial history of pheochromocytoma or medullary thyroid carcinoma and to patients who had clinical or biochemical evidence suggestive of multiple endocrine neoplasia type 2 (MEN2. In our series no subject fulfil this characteristic, making MEN2 highly unlikely in this population"

In tables: put points not commas

  • Thank you for the suggestion. We have corrected the tables by replacing commas with points as decimal separators

Results are well presented, somehow long. some sentences are summarized in tables and figures and could be deleted.

  • Thank you for your feedback. We had already aimed to keep the Results concise and focused on the essentials. Nevertheless, we have further shortened a somewhat lengthy section (lines 233-240) to improve clarity and readability

Put limitations / strengths at the end of discussion.

  • Thank you for the suggestion. We have moved the limitations and strengths of the study to the end of the Discussion as requested, lines 445-463

clinical implications must be added , correlated with current guidelines.

  • Thank you for the comment. We acknowledge that the original clinical implications section was brief. We have expanded this part in the Discussion, incorporating references to current guidelines as well as to findings from another research group. (lines 362-371)

Add a cost limitation of the procedure.

  • Thank you for the suggestion. The paragraph addressing the cost limitation of the procedure was added following your indication and is included at the end of the clinical implications section (lines 374-381)

Hypercalciuria was high in patients over 50 yo : this suggest an age-related modulation of CASR function, or  due to longer disease exposure?

  • We primarily attributed the higher prevalence of hypercalciuria in patients over 50 years to the longer duration of disease exposure that could contribute to this phenomenon. Nevertheless, a mere statistical influence of the larger sample size within this subgroup, should not be excluded. However, we agree that your observation provides a valuable perspective, and this issue merit further investigation in future studies specifically designed to explore the interplay between age, CASR activity, and clinical manifestations of PHPT.

The study did not find a link between CASR polymorphisms and renal calculi, bone disease, or aggressive histology. Please better explain in discussion if this imply that these variants have a minimal role in PHPT severity, or was the sample size too small ?

  • Thank you for this important comment. We have addressed this point by adding a dedicated paragraph in the Discussion section, where we consider whether the lack of association suggests a minimal role of CASR polymorphisms in PHPT severity or reflects limitations related to sample size. (lines 434-444)

Are there differences in response to medical therapy : cinacalcet; based on CASR genotype?

  • We thank you for raising this important point. However, the use of medical therapy, including cinacalcet, was not within the scope of the present study, and data regarding pharmacological treatment and response were not included. There is some evidence in literature addressing a different cinacalcet response in patient with CASR polymorphisms. These observation, however, are collected within a setting of secondary hyrparathyroidism. We agree that exploring the relationship between CASR genotype and response to calcimimetic therapy represents a valuable area for future research."

references are not numbered in the end.

  • We thank you for this remark. We fixed every mistake in reference numbering and reporting. Right now, the references appear to be properly numbered and formatted according to the journal’s style. We kindly ask to let us know if further modifications will be required.

Reviewer 2 Report

Comments and Suggestions for Authors

The manuscript entitled "CaSR gene polymorphisms and PHPT phenotypes: what else can 
we learn? A single-center experience with a cohort of Italian patients" . The work investigates the role of CASR gene polymorphisms (A986S, R990G, 21Q1011E) in PHPT genetic susceptibility and its clinical variability. The aim was to evaluate the prevalence of these polymorphisms in patients with sporadic PHPT and their impact on clinical course, biochemistry, and histological features. The manuscript is generally well designed and written. However, many technical errors are present. Figures should be improved for better readability. References are not cited nor listed in the appropriate way. Please correct. 

Comments on the Quality of English Language

English needs minor polishing.

Author Response

The manuscript entitled "CaSR gene polymorphisms and PHPT phenotypes: what else can 
we learn? A single-center experience with a cohort of Italian patients". The work investigates the role of CASR gene polymorphisms (A986S, R990G, 21Q1011E) in PHPT genetic susceptibility and its clinical variability. The aim was to evaluate the prevalence of these polymorphisms in patients with sporadic PHPT and their impact on clinical course, biochemistry, and histological features. The manuscript is generally well designed and written. However, many technical errors are present. Figures should be improved for better readability. References are not cited nor listed in the appropriate way. Please correct. 

  • We sincerely thank the reviewer for the constructive feedback and for acknowledging the overall design and writing quality of our manuscript. In response to the comments, we have thoroughly revised all figures to enhance their clarity and readability. Additionally, we have meticulously corrected all references to ensure they are cited and listed according to the journal’s guidelines. We believe these revisions have significantly improved the manuscript and trust it now meets the expected standards.